# *pyDHM*: A Python library for applications in digital holographic microscopy

**Raul Castañeda**[1]**, Carlos Trujillo**[2]**, Ana Doblas**[1]*

**1** Optical Imaging Research Laboratory, Department of Electrical and Computer Engineering, The University of Memphis, Memphis, TN, United States of America, **2** Applied Optics Group, School of Applied Sciences and Engineering, Universidad EAFIT, Medellin, Colombia

* adoblas@memphis.edu

**Data Availability Statement:** The pyDHM library, documentation, and the holograms used are publicly available on GitHub (https://catrujilla.github.io/pyDHM/ and https://github.com/catrujilla/pyDHM). The GitHub repository also includes

## Abstract

*pyDHM* is an open-source Python library aimed at Digital Holographic Microscopy (DHM) applications. The *pyDHM* is a user-friendly library written in the robust programming language of Python that provides a set of numerical processing algorithms for reconstructing amplitude and phase images for a broad range of optical DHM configurations. The *pyDHM* implements phase-shifting approaches for in-line and slightly off-axis systems and enables phase compensation for telecentric and non-telecentric systems. In addition, *pyDHM* includes three propagation algorithms for numerical focusing complex amplitude distributions in DHM and digital holography (DH) setups. We have validated the library using numerical and experimental holograms.

## 1. Introduction

Digital holographic microscopes (DHMs) have been widely applied in material and biological applications. For instance, among many biological and biomedical applications [1–4], DHM systems are used for analyzing cells and tissues [4–8], as well as disease diagnosis and screening [4, 9–11] and assessing the polarimetric properties of biomedical samples[12, 13]. In material science, DHM systems have been used to measure MEMs [14–19], detect and characterize defects [20], as well as to characterize surface topography [21–23]. The performance of DHM technologies relies heavily on computational reconstruction processing to provide trustworthy sample information. The required computational reconstruction algorithms are uniquely dependent on the optical configuration of the DHM system. An incorrect selection of the reconstruction algorithms leads to distorted and inaccurate amplitude and phase measurements. DHM systems record the interference pattern (e.g., hologram) generated between the scattered light from the sample, named object wavefront, and a known reference wavefront. DHM systems operate in off-axis, slightly off-axis, or in-line (also known as on-axis) configurations based on the interference angle. Therefore, the selection of the DHM reconstruction algorithms depends on this interference angle. For example, off-axis DHM systems enable the reconstruction of the complex amplitude distribution of an object wavefront from a single hologram since the three components of the hologram are entirely separable in the Fourier domain [24, 25]. Therefore, the reconstruction algorithm in off-axis DHM systems requires

troubleshooting guidelines for the correct use of the library and some instructional videos (https://youtu.be/h76nZM6JpXo, https://youtu.be/Z9o0ODe1IUQ, https://youtu.be/CMHbF0uoWDk, and https://youtu.be/CMHbF0uoWDk) on how to install and use the library.

**Funding:** This research was partially funded by the Vicerrectoría de Ciencia, Tecnología e Innovación from Universidad EAFIT, and National Science Foundation (NSF) grant number 2042563. The funders had no role in library design, implementation and validation, decision to publish, or preparation of the manuscript.

**Competing interests:** The authors have declared that no competing interests exist.

the spatial filtering of the sample frequencies from the hologram's spectrum. In contrast, the spectral components of a hologram in the Fourier domain overlay totally or partially in in-line and slightly off-axis DHM systems, respectively, requiring the acquisition of multiple phase-shifted holograms and the application of phase-shifting (PS) techniques [1, 26] to reconstruct the desired object information. In addition, the selection of the DHM reconstruction algorithms also depends on whether the DHM imaging system operates in telecentric or non-telecentric regimes. DHM systems operating in the telecentric regime only require the interference angle compensation in the off-axis and slightly off-axis configuration. Oppositely, non-telecentric DHM systems should compensate for the spherical phase factor recognized in DHM and associated with a non-telecentric imaging system [27–30]. Finally, the DHM configuration can operate in an image-plane configuration, meaning that in-focus DHM holograms are recorded, so there is no need to apply numerical propagations to focus the amplitude and phase images. However, if out-of-focus holograms are recorded, the user should numerically propagate the complex amplitude distribution to provide in-focus images. Among the different numerical propagation algorithms to reconstruct DHM images, the most used computational approaches in DHM are the angular spectrum for short propagation distances [31, 32] and the Fresnel transform algorithm for large propagation distances [33].

Still, each research group within the DHM community develops and implements its own computational reconstruction algorithms based on its experimental digital holography (DH) and DHM systems. Nonetheless, some research groups have developed and implemented libraries and plugins to address the need for an open-source reconstruction toolbox in DHM and DH. The main difference between DHM and DH systems is the utilization of a microscopic imaging system in the object wave. For example, in 2010, Shimobaba *et al.* developed a numerical library, e.g., the GWO library, for diffraction calculations using a Graphics Processing Unit (GPU) [34]. This GWO library enabled the numerical propagation of complex amplitude distributions using the angular spectrum and Fresnel approaches. Since the GWO library was based on the C language and, consequently, was not user-friendly, in 2012, the same authors developed a new C++ class library for diffraction and calculations using computer-generated holograms (CGHs) [35]. In 2015, Piedrahita-Quintero *et al.* developed a JAVA plugin for numerical wavefields propagation [36]. This plugin enabled the propagation of complex amplitude distributions using angular spectrum, Fresnel, and Fresnel–Bluestein approaches. The pluggin's most important feature is that it is embedded within the well-known open-source software for image processing named ImageJ. Piedrahita-Quintero *et al.* validated their plugging by numerically propagating experimental holograms recorded in off-axis DH and DHM systems. In 2017, the same authors upgraded their previously-developed JAVA-based plugin to a GPU-accelerated library, JDiffraction [37]. The OpenHolo library, developed by Hong *et al.* in 2020, can generate holograms using the most popular CGH algorithms [38]. This library also includes standard tools for holography, like phase unwrapping algorithms and reconstruction algorithms using the Rayleigh-Sommerfeld diffraction integral and the numerical Fresnel propagation algorithm. That same year, Trujillo *et al.* developed an ImageJ-based open-source plugin for in-line digital lensless holographic microscopy (DLHM) [39]. This plugin contains two modules to simulate holograms using the discrete version of the Rayleigh–Somerfield diffraction formula and to reconstruct holograms using the Kirchhoff–Helmholtz diffraction integral. More recently, the Manoharan Lab at Harvard University has implemented the HoloPy library, a Python-based library, to perform scattering and optical propagation theories [40]. Like the JAVA plugin from Trujillo *et al.* [39], the HoloPy library focuses on in-line DLHM systems.

Despite all these efforts, a library containing the needed computational reconstruction approaches to reconstruct DHM images, regardless of the optical configuration of the system,

does not still exist. This paper presents a Python library focused on DHM applications, named *pyDHM*, for reconstructing DHM images for various experimental DHM implementations. We aim to provide the DH and DHM community with a complete set of tools for holographic processing in a widely supported and easy-to-use programming language. The library consists of four packages. The first package contains a set of useful functions, such as calculating the hologram spectrum and displaying the amplitude and phase images. The second package is related to reconstructing in-line and slightly off-axis DHM systems using phase-shifting techniques. The third package reconstructs phase images from off-axis DHM holograms using telecentric and non-telecentric configurations. Finally, the last package includes algorithms for propagating the complex amplitude distribution using the angular spectrum and the Fresnel and Fresnel-Bluestein transform approaches. Our proposed library has been validated with experimental and simulated holograms recorded using different setups.

## 2. Background: Reconstruction algorithms for different optical DHM configurations

DHM systems are based on optical interferometry, which involves the digital recording of the interference pattern (e.g., digital hologram) between the complex amplitude distribution scattered by a microscopic object (e.g., the object wavefront) and a uniform reference wavefront. DHM systems are colloquially known as hybrid imaging systems since the final reconstructed amplitude or phase images are obtained after further computational processing of the digital hologram. This second stage, known as the reconstruction stage, depends on the optical configuration of the DHM system. This section thoroughly describes different DHM configurations and their corresponding numerical reconstruction algorithms.

For simplicity, we have chosen a traditional DHM setup based on a Mach-Zehnder interferometer operating in transmission mode, see Fig 1(A). The light generated from a laser of wavelength λ is collimated by a converging lens (CL) and the resultant plane wavefront impinges a cubic beam splitter (BS1) to generate the object (O) and the reference (R) wavefronts. A microscopic imaging system is inserted in the object arm of the DHM system. This microscopic imaging system comprises an infinity-corrected microscopic objective (MO) lens and a tube lens (TL) which forms an image of the wavefront scattered by the microscopic sample with amplitude distribution $o$ ($x,y$). If the object is set at the front-focal plane (FFP) of the MO lens, the image of the sample whose amplitude distribution is $u_{IP}$ ($x,y$) is then obtained at the back-focal plane (BFP) of the TL. This plane is known as the microscope's image plane (IP). The complex amplitude distribution $u_{IP}$ ($x,y$) produced by the microscope at the IP is given by

$$u_{IP}(x,y) = \frac{1}{M^2} e^{ik(2f_{MO}+d+f_{TL})} \exp\left(i\frac{k}{2C}(x^2+y^2)\right) \times \left[o\left(\frac{x}{M},\frac{y}{M}\right) \otimes_2 P\left(\frac{x}{\lambda f_{TL}},\frac{y}{\lambda f_{TL}}\right)\right], \quad (1)$$

where $k = 2\pi/\lambda$ is the illumination wavenumber, $\otimes_2$ denotes the 2D convolution operator, $f_{MO}$ is the focal length of the MO lens, $f_{TL}$ is the focal length of the TL, $d$ is the distance between the BFP of the MO and TL lenses (Fig 1), and $M = -f_{TL}/f_{MO}$ stands for the lateral magnification of the imaging system which does not depend on the distance $d$. $P(u,v)$ is the 2D Fourier transform of the amplitude transmittance of the pupil distribution, $p(x,y)$. The quadratic phase factor $\exp\left(i\frac{k}{2C}(x^2+y^2)\right)$ with a radius of curvature $C = f_{TL}^2/f_{TL} - d$ is associated with a non-telecentric geometry ($d \neq f_{TL}$) for the optical microscope. When the microscope operates in the telecentric regime ($d = f_{TL}$), no quadratic phase factor appears in Eq (1).

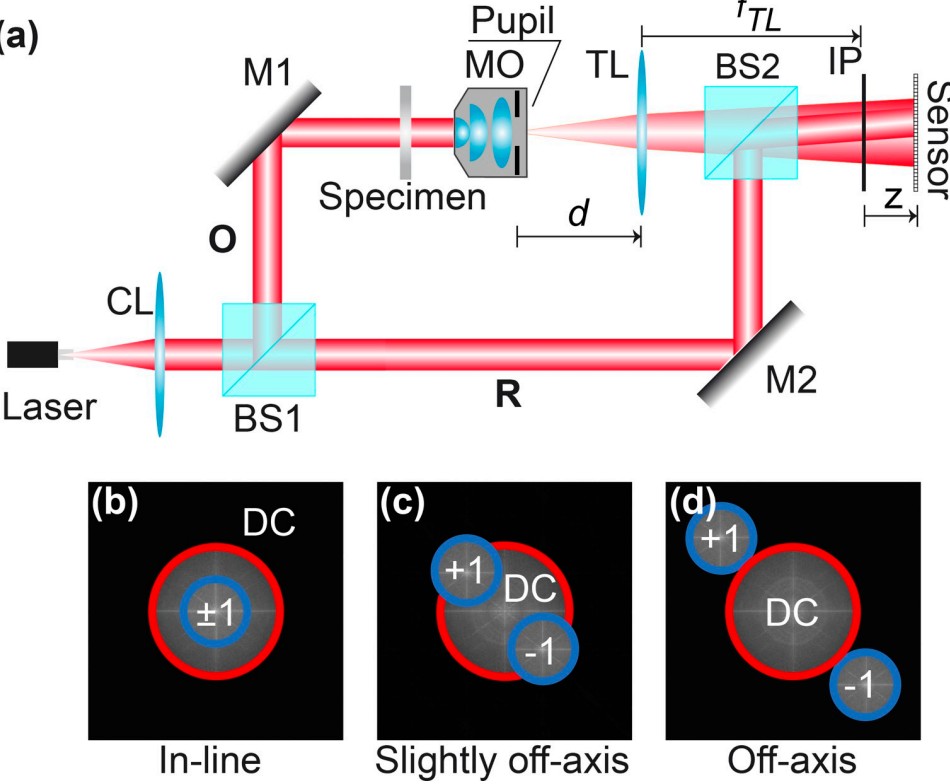

**Fig 1. Scheme of a transmission DHM.** (a) Optical setup based on a Mach-Zehnder interferometer in which the microscope objective (MO) lens and tube lens (TL) are generally arranged in non-telecentric mode (d ≠ $f_{TL}$). (b)-(d) Classification of the DHM system based on the interference angle between the object and reference waves (in-line, slightly off-axis, and off-axis) based on the Fourier spectrum of the hologram. The size of the DC diffraction order (red circle) is always double the ±1 terms (blue circles) if the DHM system operates in the telecentric regime (d = $f_{TL}$). The system components are: BS1 and BS2, beam-splitters; CL, converging lens; IP, image plane; M1 and M2 mirrors; O, object wavefront; R, reference wavefront.

The hologram, captured at any plane from the IP, is the result of the interference between the complex wavefield produced by the microscope at a distance $z$ from the IP,

$$u(x, y; z) = \frac{i}{\lambda z} e^{ikz} \left\{ u_{IP}(x, y) \otimes_2 \exp\left( i\frac{k}{2z}(x^2 + y^2) \right) \right\}, \quad (2)$$

and a tilted plane wavefront,

$$r(x, y) = \sqrt{I_R} \exp[ik(\sin\theta_x \cdot x + \sin\theta_y \cdot y)], \quad (3)$$

where $I_R$ is the irradiance of the reference wavefront, and $\boldsymbol{\theta} = (\theta_x, \theta_y)$ is the vector representation of the titled reference angle to the optical axis, which coincides with the center of the object wavefront. In Mach-Zehnder-based DHM systems, this angle can be changed by tilting the optical elements that reflect the reference wavefront $R$ (e.g., the BS2 and/or M2 in Fig 1 (A)). The irradiance distribution of the hologram $h(x,y;z)$ is

$$h(x, y; z) = |u(x, y; z)|^2 + |r(x, y)|^2 + u(x, y; z)r^*(x, y) + u^*(x, y; z)r(x, y), \quad (4)$$

where $z < 0$ refers to planes located in front of the IP, $|\cdot|^2$ represents the square module, and the symbol $^*$ stands for the complex conjugate operation. In Eq (4), the first two terms are

related to the irradiance of both object and reference wavefronts. On the other hand, the third and fourth terms in Eq (4) encode the complex amplitude information of the object wavefront scattered by the sample. These terms represent the object's real and twin images. From Eq (4), it is clear that the object information $o(x,y)$, encoded in $u(x,y;z)$ is mixed with other undesired terms. The reconstruction algorithms aim to separate these undesired terms from the object information to provide well-contrast amplitude and phase images from the complex in-focus amplitude distribution $u_{IP}(x,y) = u(x,y;z = 0)$ with minimum distortions. Assuming that the reference wavefront is plane, the 2D Fourier transform of the hologram $H(u,v;z)$ is

$$H(u, v; z) = DC(u, v; z) + U\left(u - \frac{\sin\theta_x}{\lambda}, v - \frac{\sin\theta_y}{\lambda}; z\right) + U^*\left(u + \frac{\sin\theta_x}{\lambda}, v + \frac{\sin\theta_y}{\lambda}; z\right), \quad (5)$$

where $(u,v)$ are the transverse spatial frequencies, $DC(u, v; z) = U *_2 U^* + R *_2 R^*$ being $*_2$ the 2D cross-correlation operator. The capital letters refer to the 2D Fourier transform distributions to simplify our notation. The spatial frequencies of the DC term are always placed at the center of the hologram spectrum. However, the frequencies of the ±1 terms, respectively the $U$(·) and $U^*$(·) terms, are located symmetrically around the DC term at locations that depend on the interference angle $\boldsymbol{\theta} = (\theta_x, \theta_y)$. In other words, the different components of the hologram spectrum may overlap based on the interference angle between the two wavefronts of the DHM system. Knowledge of the hologram's spectral composition is critical to selecting the reconstruction method. For example, off-axis DHM systems are those in which the angle between the object and reference wavefronts is such that the terms in Eq (5) are not superimposed [Fig 1(D)]. The reconstruction method for off-axis DHM systems involves a spatial filtering approach using a single hologram [24, 25]. The other extreme case occurs when no interference fringes are observed in the hologram since the interference angle between wavefronts is zero. Therefore, the three terms in Eq (5) entirely overlap [Fig 1(B)]. These DHM systems operate in in-line (or on-axis) regime. The third DHM configuration, slightly off-axis DHM systems, lies between these two extremes; the interference angle between both wavefronts is not null but small enough to produce some overlapping between the different components of the hologram spectrum [Fig 1(C)]. For in-line and slight off-axis DHM systems, the reconstruction algorithms involve phase-shifting (PS) techniques, requiring the recording of multiple holograms in which the phase of the reference wavefront is shifted (e.g., phase-shifted holograms) [1, 26]. The main advantage of PS algorithms is that the reconstructed phase images are obtained via point-wise subtractions and division operations between the recorded phase-shifted holograms. Traditionally, PS algorithms are aimed exclusively at strictly in-line DHM systems. We highlight the traditional PS algorithms requiring five, four, and three phase-shifted holograms among the different PS algorithms. In the five- and four-step algorithms, the phase shift between the holograms is π/2. Consequently, the point-wise phase images are reconstructed by

$$\varphi(x, y) = \tan^{-1}\left(\frac{2[h(x, y; 3\pi/2) - h(x, y; \pi/2)]}{2h(x, y; \pi) - h(x, y; 0) - h(x, y; 2\pi)}\right), \quad (6)$$

and

$$\varphi(x, y) = \tan^{-1}\left(\frac{h(x, y; 3\pi/2) - h(x, y; \pi/2)}{h(x, y; \pi) - h(x, y; 0)}\right), \quad (7)$$

for the five- and four-step algorithms, respectively. The third variable of the hologram distribution in these equations refers to the phase shift of the reference wavefront. For example, $h(x, y;3\pi/2)$ is a recorded hologram in which there is a phase shift of 3π/2 to the first hologram.

One can reconstruct the phase distribution using a three-step PS algorithm with a phase shift of $2\pi/3$ between holograms as

$$\varphi(x,y) = \tan^{-1}\left(\sqrt{3}\,\frac{h(x,y;\pi/3) - h(x,y;5\pi/3)}{h(x,y;5\pi/3) + h(x,y;\pi/3) - 2h(x,y;\pi)}\right). \qquad (8)$$

Although the three-step PS algorithm requires fewer holograms, being more suitable for real-time DHM imaging, this implementation is more sensitive to noise than the four- and five-step PS algorithms in experimental conditions. For this reason, four- and five-step algorithms are still used in many phase-shifting DHM configurations.

Due to the experimental difficulty in perfectly aligning the object and reference wavefronts, thus achieving a strictly in-line setup, slightly off-axis DHM systems are commonly preferred. For slightly off-axis DHM systems, traditional PS algorithms must be modified to compensate for the interference angle between both interfering wavefronts. De Nicola *et al.* proposed a four-step PS strategy with a phase shift of $\pi/2$ between consecutive holograms [41]. In this quadrature method, the complex amplitude distribution of the object can be reconstructed by summing the individual products between the recorded holograms and their corresponding digital reference wavefronts

$$\begin{aligned}\hat{u}(x,y) = {}& h(x,y;0)\hat{r}(x,y;0) + h(x,y;\pi/2)\hat{r}(x,y;\pi/2) \\ & + h(x,y;\pi)\hat{r}(x,y;\pi) + h(x,y;3\pi/2)\hat{r}(x,y;3\pi/2).\end{aligned} \qquad (9)$$

Eq (9) provides the complex amplitude distribution of the specimen without the DC term and the conjugate image, enabling the computing of amplitude and phase images via $|\hat{u}(x,y)|$ or $\mathrm{atan}(\mathrm{imag}[\hat{u}(x,y)], \mathrm{real}[\hat{u}(x,y)])$, respectively.

The main disadvantage of traditional in-line and slightly off-axis PS approaches is that most of these methods require i) accurate knowledge of the phase shift between the recorded holograms, and ii) this phase shift must be equal within the acquisition sequence. These two requirements can be experimentally challenging due to inaccuracies prevalent in most phase-shifting devices, particularly those based on mechanical movements. Several blind PS strategies have been proposed to have multiple holograms with a random and unknown phase shift [42–49]. Among these approaches, we have proposed two blind iterative PS-DHM algorithms for slightly off-axis DHM systems, which are computationally efficient, user-friendly, and robust under noisy conditions [50, 51]. These two blind PS-DHM methods are based on the demodulation of the terms composing the Fourier spectrum of the hologram. In particular, the hologram distribution can be understood as a linear combination between three unknown components $\{d_0, d_{+1}$ and $d_{-1}\}$ where the weighting of each component depends on the phase shift: $h = d_0 + e^{-\mathbf{j\Delta\theta}}\,d_{+1} + e^{\mathbf{j\Delta\theta}}\,d_{-1}$ [50] where $d_0$ is the two first terms of Eq (4), $d_{+1} = u(\mathbf{x},y;z)e^{-jk\sin\theta\cdot x}$, and $d_{-1} = u^*(x,y;z)e^{jk\sin\theta\cdot x}$, being $\mathbf{x} = (x,y)$ the vector representation of the lateral coordinates. Based on this composition of the hologram distribution in terms of $\{d_0, d_{+1},$ and $d_{-1}\}$, the object information can be reconstructed from either the $d_{+1}$ or $d_{-1}$ components using three recorded holograms, $\{h_1, h_2, h_3\}$ with different phase shifts $\{\Delta\theta_1, \Delta\theta_2, \Delta\theta_3\}$. Conversely, the hologram can also be written as the sum of two components $\{d_0, d_3\}$: $h = d_0 + e^{\mathbf{j\Delta\theta}}\,d_3$, where $d_3 = d_{+1} + e^{-\mathbf{j2\Delta\theta}}\,d_{-1}$, requiring only two holograms with a phase-shift difference of $\Delta\theta$ to estimate $d_3$ [51]. An accurate estimation of the $d_{+1}$ or $d_3$ components involves that the phase shifts used to demodulate the $d_{+1}$ or $d_3$ components coincide to the ones of the experimental reference wave. When the values of the phase shifts match the real ones, the spectrum of the $d_0$ component is composed of a unique order for the 2- and 3-step blind PS algorithms. The spectrum of the $d_{+1}$ component is also composed of a unique order for the 3-step blind PS algorithm. Our blind PS-DHM approaches take advantage of this observation, e.g.,

the expected/real composition of the spectrum of the decomposed terms, to estimate jointly the phase steps and the $d_{+1}$ or $d_3$ component by minimizing a cost function that quantifies the difference between the absolute value of the Fourier component of the expected/real and residual peaks. Since both algorithms are used for slightly off-axis DHM systems, one must compensate for the linear phase term introduced by the tilted plane reference wave. In addition, the 2-step blind PS algorithm requires the spatial filtering of the spectral frequencies of the $d_{+1}$ term from the spectrum of the demodulated $d_3$ term. Consequently, the 2-step blind PS approach only works for slightly off-axis DHM systems in which the spectra of the $d_{+1}$ and $d_{-1}$ terms do not overlap in the Fourier domain among them. The major limitation of these approaches is that they only work for DHM systems operating in telecentric regime since the center of the ±1 terms in the Fourier transform should correspond to a maximum value.

The reconstruction algorithms in in-line and slightly off-axis DHM systems using PS strategies require multiple recorded holograms, restricting the use of those systems for live-cell imaging and dynamic analysis. Therefore, DHM systems operating in an off-axis regime are the most used DHM system for real-time imaging. These systems allow the reconstruction of an object's amplitude and phase information from a single recorded hologram. The reconstruction algorithms for reconstructing a phase image in off-axis DHM systems involve two steps. The first step is the spatial filtering of the frequencies related to the object from the hologram spectrum. Due to the off-axis configuration, the ±1 diffraction orders are arranged symmetrically around the DC term in the Fourier space [Fig 1(D)]. From the hologram spectrum [Eq (5)], the spectral object information (i.e., the +1 term) can be filtered [24].

$$H_F(\boldsymbol{u}) = U\left(u - \frac{\sin\theta_x}{\lambda}, v - \frac{\sin\theta_y}{\lambda}; z\right). \qquad (10)$$

Eq (10) represents the filtered hologram spectrum, which is the spectrum of the sample displaced at the spatial frequencies ($\sin\theta_x/\lambda$, $\sin\theta_y/\lambda$). The amplitude distribution scattered by the sample can be obtained as the absolute value of the inverse Fourier transform of Eq (10). Whereas the spatial filtering step is sufficient for amplitude imaging, quantitative phase imaging requires the phase compensation of the tilting angle between the object and reference wavefronts (the second step of the reconstruction procedure). The phase compensation of the interference angle can be performed in the space domain by multiplying the inverse Fourier transform of Eq (10), $h_F(\boldsymbol{x})$ and a digital replica of the reference wavefront, $r_D(\boldsymbol{x})$ as $\hat{u}(\boldsymbol{x}) = r_D(\boldsymbol{x}) \cdot h_F(\boldsymbol{x})$. Assuming a uniform plane reference wavefront, the generation of a digital reference wavefront requires the knowledge of the interference angle $\boldsymbol{\theta} = (\theta_x, \theta_y)$, which depends on the wavelength of the light source, the features of the digital sensor (i.e., $M \times N$ square pixels with pixel size $\Delta_{xy}$), and the subtraction between the pixel locations of the DC and the +1 terms in the hologram spectrum [52]. Among these parameters, the only one that is unknown *a-priori* is the location of the +1 term. The determination of this parameter must be precisely executed to provide phase images without phase nuisances. Although this step can be performed manually by generating a set of reconstructed phase images with different phase compensation until the one without sawtooth fringes is obtained, several automated DHM reconstruction approaches [52–56] have been proposed to alleviate this task. Among these works, the *pyDHM* library implements the following three algorithms that are restricted to off-axis DHM systems operating in telecentric regime. The first two algorithms have been coined the full region of interest (ROI) search [53] and efficient ROI search [56] from Trujillo's laboratory. The third approach has been investigated and developed by Doblas' laboratory [52]. The main difference between these three algorithms relies on how the search for the optimal parameters of the digital reference wavefront is performed. In Ref. [53], this search is carried

out using nested loops in which different spatial frequencies around the point of maximum energy of the +1 diffraction order (region of interest) are used to generate different compensating angles. Then, different digital reference wavefronts are built with these angles to produce different compensated phase maps of the object. The parameters of the reference wavefront yielding the best-compensated phase image are the proposal's output. A quantification of the number of phase discontinuities, performed via a summation-and-thresholding metric, is employed to determine the best-compensated phase image in Ref. [53]. More efficiently, in Ref. [56], the search involves a heuristic determination of the spatial frequencies inside the region of interest most likely to provide properly compensated phase maps. The latter path-oriented strategy uses only the resulting best-compensated phase map in each fewer-step nested loop until the same metric to quantify phase discontinuities as in Ref. [53] can not be further improved. Finally, in Ref. [52], the search of the parameters of the digital reference wavefront is based on an unconstrained nonlinear optimization procedure involving the minimization of the inverse of the summation-and-thresholding metric as a function of spatial frequencies around the point of maximum energy of the +1 diffraction order.

In addition to accurately determining the interfering angle between the object and reference wavefronts, the reconstruction algorithms depend on the microscope's optical configuration. The shape of the ±1 diffraction orders changes between DHM systems operating in telecentric and non-telecentric configurations, depending on the radius of the curvature of the spherical wavefront in Eq (1), $C^{-1}$. The smaller the radius of curvature $C$, the wider the ±1 orders, being the area of the ±1 diffraction orders inversely proportional to $C$ [57]. Fig 2 illustrates the changes in the size of the ±1 diffraction orders based on the radius of curvature $C$. The ±1 diffraction orders are rectangular-based compact support functions in non-telecentric DHM systems. However, these terms are circular compact support functions in telecentric-based DHM systems whose diameter is related to the resolution of the microscopic imaging

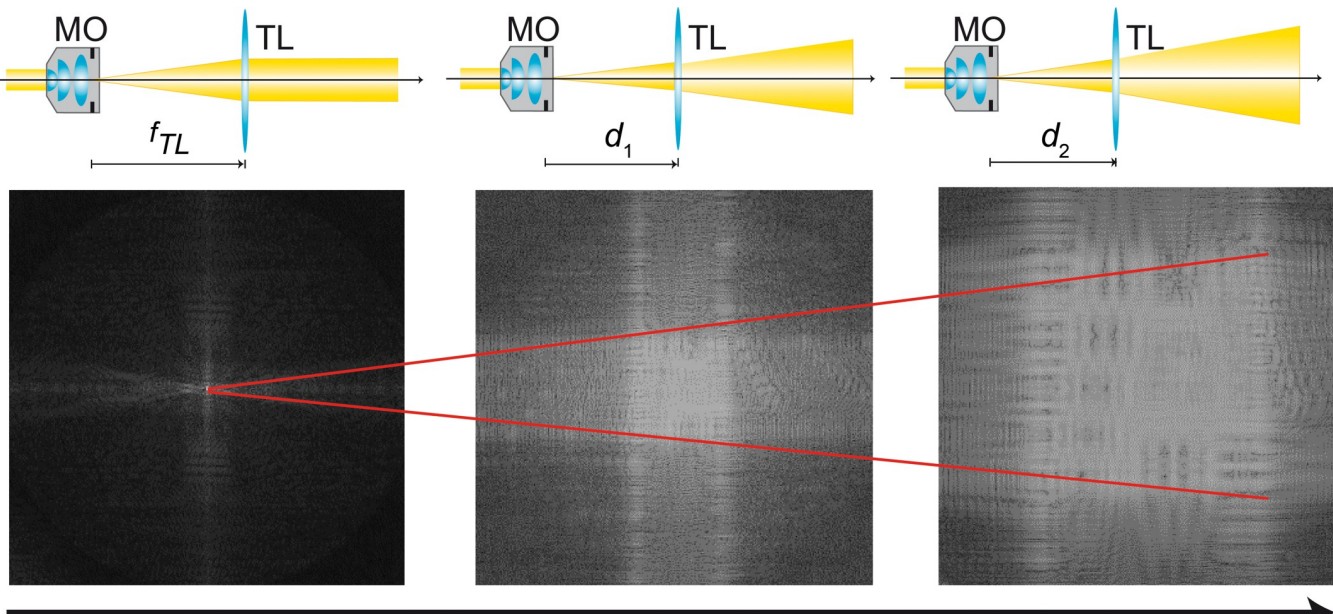

**Increasing of the size of the ±1 term by changing the distance between the MO and TL lenses**

**Fig 2. Spreading of the ±1 diffraction orders when the DHM system does not operate in telecentric regime.**

system [57], $u_c = $ NA/($\lambda M$) where NA is the numerical aperture of the MO lens. The reconstructed phase image from a non-telecentric DHM system is distorted by the quadratic phase factor that appears in Eq (1). Therefore, this quadratic phase factor should be suppressed to reconstruct accurate quantitative phase images in non-telencetric DHM systems. This quadratic phase factor can be suppressed computationally by a point-wise subtraction of the reconstructed phase with and without a sample, e.g., performing a double-exposure technique where two holograms should be recorded [58]. Single-shot computational approaches have also been proposed to eliminate the quadratic phase factor by multiplying the inverse Fourier transform of Eq (10), $h_F(\boldsymbol{x})$, with the conjugated replica of the distorted phase term, $\exp\left[-\mathrm{i}\frac{2\pi}{\lambda C}\left[(x - x_C)^2 + (y - y_C)^2\right]\right]$ knowing its center ($x_C$, $y_C$) and radius of curvature ($C$). These parameters can be estimated by analyzing the hologram's spectrum [59].

Finally, numerical propagators of complex wavefields are required in DHM if digital holograms are not recorded at the IP since the object's reconstructed information must be numerically focused considering the axial distance $z$ between the sensor/hologram plane and the in-focus plane [60]. Conventional numerical propagators are based on the angular spectrum or Fresnel Transform approaches [31, 32]. The angular spectrum approach represents a complex amplitude wavefront as a combination of infinite plane wavefronts following Huygens' principle. Therefore, the in-focus complex amplitude distribution can be estimated as

$$\hat{u}_{IP}(x, y) = \mathrm{FT}^{-1}\left[\hat{U}(u, v; z) \bullet \exp\left(-\mathrm{i}\frac{2\pi}{\lambda}z\sqrt{1 - \lambda^2(u^2 + v^2)}\right)\right], \tag{11}$$

where $\mathrm{FT}^{-1}[\cdot]$ denotes the inverse 2D Fourier transform operator, and $\hat{U}(x, y; z)$ is the 2D Fourier transform of the reconstructed complex amplitude distribution [$\hat{u}(x, y; z)$] after applying PS algorithms or spatial filtering and phase compensation the interfering angle. The phase map of the $\hat{u}(x, y; z)$ distribution should have been compensated for any quadratic phase factor. The Fresnel Transform approach is another method to solve the Fresnel-Kirchhoff diffraction equation using the paraxial approximation [32]. The paraxial approximation involves that the sensor/hologram plane dimensions are smaller than the propagation distance, i.e., large propagation distances. Based on the Fresnel Transform, the relationship between the out-of-focus distribution, $\hat{u}(\boldsymbol{x}, \boldsymbol{y}; \boldsymbol{z})$, and the in-focus complex amplitude distribution, $\hat{u}_{IP}(x, y)$, is expressed as

$$\hat{u}_{IP}(x, y) = \frac{-\mathrm{i}}{2\lambda}\iint \hat{u}(x_0, y_0; z)\exp\left(-\mathrm{i}\frac{\pi}{\lambda z}(x^2 + y^2)\right)\exp\left(\mathrm{i}\frac{2\pi}{z}(xx_0 + yy_0)\right)\mathrm{d}x_0\mathrm{d}y_0. \tag{12}$$

Some constant phase factors have been neglected in Eq (12). The third term within the integral in Eq (12) is related to the kernel of a 2D Fourier transform with frequencies $u = x/z$ and $v = y/z$. Although the low-computational complexity of the Fresnel transform enables fast numerical processing, this traditional approach imposes a fixed magnification of the propagated wavefield according to the illumination wavelength and the system's geometry. A third numerical propagator can be employed to overcome this limitation. In a modified version of the Fresnel approach, the kernel of the Fourier transform in Eq (12) is modified by inserting the Bluestein substitution [33, 61] to convert this expression into a convolution operation in which the magnification of the propagated wavefield can be chosen at will at the expense of higher computational complexity.

In summary, the DHM numerical processing involves computational strategies dependent on the employed optical configuration. In the proposed *pyDHM* library, we have implemented functions for: i) spatial filtering off-axis holograms, ii) computing complex amplitudes distributions from phase-shifted interferograms in in-line and slightly off-axis setups, iii)

compensating the interference angle between interfering wavefronts in off-axis setups, iv) compensating the spherical wavefront of non-telecentric recordings, and v) the propagating complex wavefields to focus non-IP holographic reconstructions.

## 3. Library structure

The *pyDHM* library consists of four packages. The first utility package includes essential functions such as reading, displaying images, and filtering Fourier spectrums of holograms. The second package is related to reconstructing in-line and slightly off-axis DHM systems using phase-shifting techniques. The third package reconstructs phase images from off-axis telecentric-based DHM holograms. Finally, the last package includes algorithms for propagating complex amplitude distributions using the angular spectrum and the Fresnel and Fresnel-Bluestein transform approaches. This section explains each package in detail, including the functions and required parameters. We also present sample codes and results illustrating the performance of the *pyDHM* library.

### 3.1 Package 1: Utility package

The first package in the *pyDHM* library contains functions for reading and displaying images, computing the Fourier transform (FT), and applying filters to reduce speckle noise [62]. Since the library focuses on DHM applications dealing with complex amplitude distributions, one can display any complex wavefield's amplitude, intensity, or phase map. Although these operations can be straightforwardly implemented in Python for experienced users, this package is aimed to provide compact and user-friendly codes. This package is imported by typing the following code lines, *from pyDHM import utilities*. Table 1 shows the information for each package function, including the declaration statement and the parameters needed. Examples of the use of this package are shown in the upcoming figures.

### 3.2 Package 2: Phase-shifting package

The second package in the *pyDHM* library contains the phase-shifting strategies for reconstructing the complex amplitude distribution in in-line and slightly off-axis systems: the following code line, *from pyDHM import phaseShifting*, calls this package. The package is composed of six different phase-shifting approaches. We have implemented the traditional phase-shifting techniques in which the phase shifts are known using 5 (PS5), 4 (PS4) and 3 (PS3) phase-shifted images, which corresponds to Eqs (6)–(8), respectively. We have also implemented the quadrature PS method (SOSR) [41] and two blind PS approaches using 3 (BPS3) and 2 (BPS2) frames [50, 51] for slightly off-axis DHM systems. The two blind PS approaches require a DHM operating in telecentric regime. Table 2 illustrates the different PS strategies implemented in the package, their definition line statement, and respective parameters. These PS functions require the input of multiple phase-shifted holograms. The number of holograms and the phase shift values depend on the phase-shifting strategy used. For example, 5 holograms (e.g., inp0, inp1, inp2, inp3, inp4) are required with phase shift of $\pi/2$ (e.g., 0, $\pi/2$, $\pi$, $3\pi/2$, $2\pi$) for the PS5 function. The phase shift in the PS4 algorithm is the same as the one from PS5. However, for the PS3 function, the phase shift is equal to $2\pi/3$. The SOSR strategy is based on the quadrature phase-shifting method proposed by De Nicola *et. al* [41]. The main difference between the original SOSR strategy and our implementation is that we have upgraded the method to automatically calculate the best digital reference wavefront to reconstruct fully compensated phase maps. The best digital reference wavefront is found using the ROI search [53]. The SOSR function requires 10 input parameters: 4 phase-shifted holograms with a phase shift of $\pi/2$; a True/False Boolean variable, *upper*, which is related to the position

**Table 1. Available functions in the utility package.**

| Utility function | Specifications |
|---|---|
| **imageRead** | imageRead(namefile) |
| | Function to read an image. The parameter *namefile* corresponds to the name of the image to be opened (e.g., the hologram). |
| **imageShow** | imageShow(inp, name) |
| | Function to display an image. Two parameters are necessary: *inp* is the data to be visualized (e.g., the load hologram, the amplitude, intensity or phase distribution), and a *name* is a label for the displayed image. |
| **amplitude** | amplitude(output, log) |
| | Function to compute the amplitude distribution of the output complex wavefield. Two parameters are necessary: *output* is the complex wavefield distribution, and *log* corresponds to a Boolean variable (e.g., True or False) for applying a common logarithm transformation to the amplitude distribution. |
| **intensity** | intensity(output, log) |
| | Function to compute the intensity distribution of the output complex amplitude wavefield. Two parameters are necessary: *output* is the complex wavefield distribution, and *log* is the Boolean variable to apply a common logarithm transformation. |
| **phase** | phase(output) |
| | Function to compute the phase distribution of an output complex wavefield distribution. The only required parameter is the *output* complex wavefield distribution. |
| **Fourier Transform** | FT(input) |
| | Function to compute the 2D Fourier transform of an image. The only required parameter is the image (*input*). |
| **Inverse Fourier Transform** | IFT(input) |
| | Function to compute the 2D inverse Fourier transform of a spectral image. The only required parameter is the spectral image (*input*). |
| **Circular filter** | sfc(field, radius, centX, centY, display) |
| | Function to filter the Fourier Transform of a hologram using a circular mask. The required parameters are: *field* the hologram, *radius* is the radius of the circular mask in pixels, and (*centX, centY*) are the central pixel positions for the circular mask. The Boolean parameter *display* shows the filtered object frequencies from the hologram spectrum when its value is true. |
| **Rectangular filter** | sfr(field, x1, x2, y1, y2, display) |
| | Function to filter the Fourier Transform of a hologram using a rectangular mask. The required parameters are: *field* the hologram; (*x1, y1*) the pixel coordinates of the upper left corner for the rectangular mask; and (*x2, y2*) the pixel coordinates of the lower right corner for the rectangular mask. The Boolean parameter *display* shows the filtered object frequencies from the hologram spectrum when its value is true. |
| **Manual rectangular filter** | sfmr(field, display) |
| | Function to filter the Fourier Transform of a hologram using a user-defined rectangular mask from a popup window. The required parameters are the hologram (*field*) and the Boolean parameter (*display*) that allows the display of the filtered hologram spectrum when it is true. It is important to mention that this function works only if the OpenCV library is installed. |
| **Hybrid median-mean** | HM2F(inp, kernel) |
| | Function to apply the median-mean filter to reduce speckle noise [62]. The parameters are: *inp* the reconstructed amplitude or phase image to be applied the filter; and the *kernel* corresponds to the maximum kernel size for the median filter. |

of the spectrum of the real image in the Fourier domain; *wavelength* is the wavelength of the illumination source used to record the hologram; *dx* and *dy* are the pixel sizes for both the input and output planes along the x- and y- directions; and *s* and *step* are two parameters for determining the ROI in the spatial frequencies domain for the compensation step. This ROI (centered at its brightest pixel) is gridded into a regular rectangular grid whose size in pixels is

**Table 2. Available functions in the phase-shifting package.**

| Phase-shifting strategy | Specifications |
|---|---|
| **5 frames** | PS5(inp0, inp1, inp2, inp3, inp4) |
| | Function to reconstruct the phase distribution using 5 phase-shifted holograms with a phase shift of π/2. The required parameters are the five holograms. |
| **4 frames** | PS4(inp0, inp1, inp2, inp3) |
| | Function to reconstruct the phase distribution using 4 phase-shifted holograms with a phase shift equal to π/2. The required parameters are the four holograms. |
| **3 frames** | PS3(inp0, inp1, inp2) |
| | Function to reconstruct the phase distribution using 3 phase-shifted holograms with a phase shift equal to π/3. The required parameters are the three holograms. |
| **quadrature method** | SOSR(inp0, inp1, inp2, inp3, upper, wavelength, dx, dy, s = 1, steps = 4) |
| | Function to reconstruct the phase distribution using 4 phase-shifted holograms with a phase shift equal to π/2. This method is based on the SOSR approach proposed by De Nicola *et al.* [41]. The input parameters are the four holograms, *inp0-inp3*; *upper* corresponds to a region for searching the diffraction order; *wavelength* is the illumination wavelength, (*dx,dy*) are the pixel sizes along the *x*- and *y*- axis, respectively. *s* is a parameter to determine the ROI size in pixels in each dimension to search for the best spatial frequency (size in each dimension equals *1+2s*). *steps* is the number of steps for the search inside the ROI in each dimension. |
| **Blind 3 raw frames** | BPS3(inp0, inp1, inp2, wavelength, dx, dy) |
| | Function to reconstruct the phase distribution using 3 phase-shifted holograms with an arbitrary and unknown phase shift. The input parameters are the three holograms (*inp0-inp2*); the illumination wavelength (*wavelength*), and the pixel sizes for x and y axes (*dx, dy*). This method is valid for slightly off-axis DHM systems operating in telecentric regime [50]. |
| **Blind 2 raw frames** | BPS2(inp0, inp1, wavelength, dx, dy) |
| | Function to reconstruct the phase distribution using 2 phase-shifted holograms with an arbitrary and unknown phase shift. This method is valid for slightly off-axis DHM systems operating in telecentric regime [51]. The spectral +1 and -1 components in the spectrum of the recorded hologram should not overlay. The input parameters are the two holograms (*inp0-inp1*); the wavelength of the illumination (*wavelength*), and the pixel sizes for x and y axes (*dx, dy*). |

given by *1+2s* in each dimension. The number of points inside this grid in each dimension is given by *step*; these points are placed apart equidistantly. Therefore, *(step)²* is the total number of points used inside the ROI. For instance, if using *s = 1* and *step = 4*, a *3x3* pixels ROI size with 16 points inside is selected. If using *s = 2* and *step = 4*, a 5*x*5 pixels ROI size with 16 points is selected, implying that a larger ROI with a lesser density of points will be used for the search. The user can adjust these parameters at will, considering that although a more accurate search can be performed by increasing their values, the computational complexity of the procedure increases as well. Finally, the parameters of the BPS3 and the BPS2 algorithms are the input phase-shifted holograms with arbitrary phase shift, and the wavelength of the illumination source and the pixel size along the x- and y-direction (e.g., *dx* and *dy*, respectively) to generate the digital reference wavefront and reconstruct fully-compensated phase maps. The variables (e.g., wavelength, *dx*, *dy*) in the functions should be inserted in the same units.

These six PS algorithms have been validated using simulated and experimental holograms. Fig 3 shows the code and an example for the PS5, PS4 and PS3 algorithms. The code starts importing the utility package and the PS package, lines 2–3 in Fig 3(A). Lines 5–9 are the commands to read the phase-shifted holograms. For simplicity, we do not display any of these input holograms. Line 11 calls functions to display the PS5, PS4 or PS3 implementation. Finally, line 13 is the command to display the phase distribution reconstructed by the selected algorithm. Fig 3(C) illustrates the reconstructed phase distribution from the simulated phase-

**(a) Example code for the in-line PS strategies**

```
1- # import packages
2- from pyDHM import utilities
3- from pyDHM import phaseShifting

4- #Load the holograms

5- inp0 = utilities.imageRead('holo1.jpg')
6- inp1 = utilities.imageRead('holo2.jpg')
7- inp2 = utilities.imageRead('holo3.jpg')
8- inp3 = utilities.imageRead('holo4.jpg')
9- inp4 = utilities.imageRead('holo5.jpg')

10- #Phase shifting using the PS strategies
11- output = phaseShifting.PS5(inp0,inp1,inp2,inp3,inp4)
output = phaseShifting.PS4(inp0,inp1,inp2,inp3)
output = phaseShifting.PS3(inp0,inp1,inp2)
12- #Display the phase reconstruction
13- phase = utilities.phase(output)
14- utilities.imageShow(phase, 'Phase reconstruction')
```

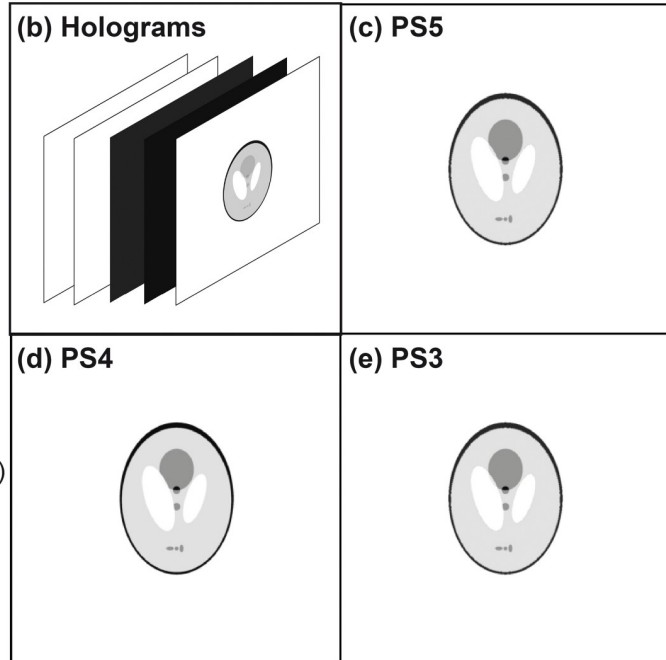

**Fig 3. Verification of the in-line PS function.** (a) An example code; (b) Simulated in-line DHM holograms of a phantom model. Panels (c)-(e) and the reconstructed phase images for the PS5, PS4, and PS3 strategies, respectively.

shifted phantom hologram [Fig 3(B)] recorded in an in-line DHM system with a phase shift of π/2. Fig 3(D) and 3(E) show the reconstructed phase images of the simulated phantom hologram using PS4 and PS3 functions, respectively.

The performance of the slightly off-axis PS strategies is evaluated using two different experimental samples. The validation of the SORS function is provided by reconstructing the phase image of a Fresnel lens with a phase shift of π/2 between holograms introduced by a liquid lens [63]. The sample used for validating the blind strategies (i.e., the BPS3 and BPS2 algorithms) is a phase USAF test target. Fig 4(A) shows the sample code to use the slightly off-axis strategy algorithms, see the Fourier spectrum of one hologram from the Fresnel lens in Fig 4(B). Again, line 10 corresponds to the command for the slightly off-axis strategies (e.g., SOSR, BPS3 and BPS2) with their corresponding parameters. Fig 4(C) to 4(E) show the reconstructed phase images of the Fresnel lens and the USAF target with minimum phase distortions.

### 3.3 Package 3: Fully-compensated phase reconstruction package

The third package of the *pyDHM* library is devoted to the phase reconstruction of DHM holograms without or with minimal perturbations (e.g., fully-compensated reconstructed phase images without distorting sawtooth fringes) using an off-axis system. One includes the *from pyDHM import phaseCompensation* line to call this package. In this package, we have implemented four functions, three functions for holograms recorded in telecentric regime: the full-ROI-search (FRS) function, the efficient-ROI -search (ERS) function, and the cost-function-search (CFS) function. And one function for holograms recorded in non-telecentric regime, the compensation non-telecentric (CNT) function. Table 3 shows the definition statement and a brief description of each package function. For example, the FRS function has seven input parameters: the off-axis hologram (*inp*), a True/False Boolean variable (*upper*) for choosing the region where the algorithm would find the maximum peak value of the +1 or -1 order for

**(a)** **Example code for the slightly off-axis strategies**

```
1- # import packages
2- from pyDHM import utilities
3- from pyDHM import phaseShifting

4- #Load the holograms
5- inp0 = utilities.imageRead('holo1.jpg')
6- inp1 = utilities.imageRead('holo2.jpg')
7- inp2 = utilities.imageRead('holo3.jpg')
8- inp3 = utilities.imageRead('holo4.jpg')

9- #Phase shifting via SOSR, BPS3 or BPS2
10- output = phaseShifting.SOSR(inp0,inp1,inp2,inp3,
            633e-9, 6.9e-6, 6.9e-6, 1, 4)
output = phaseShifting.BPS3(inp0,inp1,inp2,inp3,
            0.532, 2.9, 2.9)
output = phaseShifting.BPS2(inp0,inp1,0.532, 2.9, 2.9)

11- #Display the phase reconstruction
12- phase = utilities.phase(output)
13- utilities.imageShow(phase, 'Phase reconstruction')
```

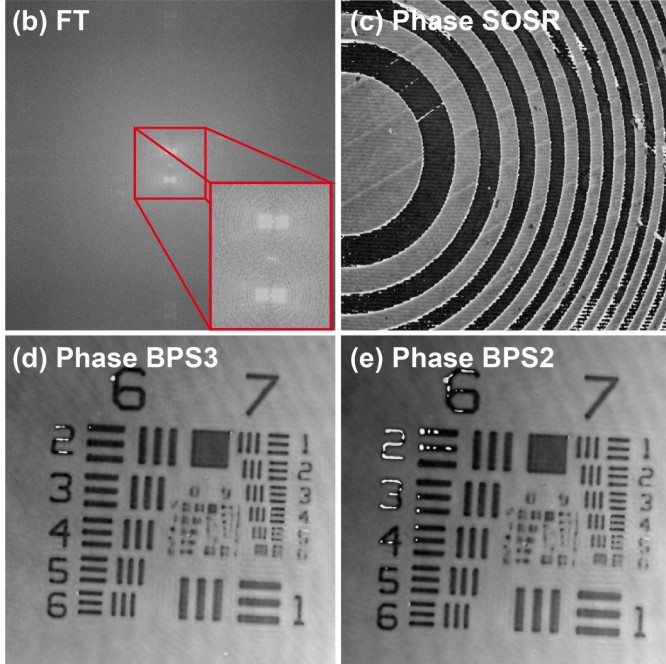

**Fig 4. Example of use of the slightly off-axis strategies.** (a) A sample code; (b) FT of recorded hologram showing that the DHM system is operating in slightly off-axis configuration. Panel (c) is the reconstructed phase image of a Fresnel lens by the SOSR function using four holograms with a π/2 phase shifting. Panels (d) and (e) are the reconstructed phase images of a USAF test target using the BPS3 and BPS2 strategies, respectively.

the filtering step. *Wavelength* corresponds to the wavelength used to record the hologram; $dx$ and $dy$ are the pixel size of the camera sensor for the acquisition of the hologram along the x- and y- direction, respectively, and $s$ and *step* are parameters for selecting the search region to find the best phase reconstructed image. These parameters determine the ROI size and the number of points inside this search region. For example, if using $s = 2$ and *step = 10*, a 2x2 pixels ROI size with 100 spatial frequency locations is selected to search for the best phase reconstructed image [53]. For using the efficient ROI search, the EFR function must be executed. This function has the same parameters as the FRS function. To run the cost-function search, the CFS function must be called. The parameters for this function are *inp*, *wavelength*, *dx*, and *dy*. By the other hand, the CNT function contain 8 parameters. Whereas the first ones are *inp*, *wavelength*, *dx*, *dy*, already defined parameters, ($x1$, $x2$, $y1$, $y2$) are the pixels position to create a rectangular mask for filtering the +1 diffraction order, where ($x1$, $y1$) and ($x2$, $y2$) are the pixel position of the upper-left and bottom-right corner, respectively.

We have reconstructed two different holograms to evaluate the performance of the fully-compensated phase reconstruction package using an off-axis DHM system operating in the telecentric regime. The *full* and *efficient ROI search* strategies have been validated using the hologram of a transverse section of the head of a *Drosophila melanogaster* fly [30]. The parameters of the DHM system to record this sample are wavelength $\lambda = 633$ nm and a camera with a pixel size $dx = dy = 6.9$ μm. We have used the hologram of a phase star target to validate the *cost-function search* strategy using $\lambda = 532$ nm, and $dx = dy = 2.6$ μm. These two holograms were recorded at the IP of the microscope. Therefore, no refocusing step is necessary for the reconstruction stage. The sample code is shown in Fig 5(A). The code starts with the import of the utility and the fully-compensated phase reconstruction packages; lines 2–3 in Fig 5(A). In line 5, the hologram is loaded. Line 7 calls the functions for reconstructing off-axis holograms in telecentric regimen: FRS, ERS, or CFS function. In Fig 5(B), the common logarithm of the

**Table 3. Available functions in the phase reconstruction package.**

| Compensation method | Specifications |
|---|---|
| **Full ROI search** | FRS(inp, upper, wavelength, dx, dy, s, step) |
| | Function to reconstruct fully-compensated phase images using the full ROI search strategy [53]. The function requires seven parameters: *inp*, *upper*, *wavelength*, *dx*, *dy*, *s*, and step. The *upper* is a True/False Boolean variable for selecting the region where the algorithm would find the maximum peak value of the +1 order for the filtering step. The default values of these parameters are $s = 2$, and *step* = 10. The DHM system should operate in off-axis configuration and telecentric regime. |
| **Efficient ROI search** | ERS(inp, upper, wavelength, dx, dy, s, step) |
| | Function to reconstruct fully-compensated phase images using the efficient ROI search strategy [56]. The function requires seven parameters: *inp*, *upper*, *wavelength*, *dx*, *dy*, *s*, and step. The *upper* is a True/False Boolean variable for selecting the region where the algorithm would find the maximum peak value of the +1 order for the filtering step. The default values of these parameters are $s = 5$, and *step* = 0.2. The DHM system should operate in off-axis configuration and telecentric regime. |
| **Cost-function search** | CFS(inp, wavelength, dx, dy) |
| | Function to reconstruct fully-compensated phase images by minimizing a cost function [52]. The function requires four parameters: *inp*, *wavelength*, *dx*, and *dy*. The DHM system should operate in off-axis configuration and telecentric regime. |
| **Compensation no-telecentric regime** | CNT(inp, wavelength, dx, dy, x1, x2, y1, y2, spatialFilter) |
| | Function to reconstruct fully-compensated phase images of holograms recorded in non-telecentric regime. The function requires up to nine parameters: *inp*, *wavelength*, *dx*, *dy*, *x1*, *x2*, *y1*, *y2*. The units of *wavelength*, *dx*, *dy* should be microns. The DHM system should operate in off-axis configuration and non-telecentric regime. The parameter spatialFilter allows to decide how to select the object frequencies from the hologram spectrum. Use spatialFilter = '*sfmr*' for a manual filter via a popup window or spatialFilter = '*sfr*' for a rectangular mask filter defined by the *x1*, *x2*, *y1*, *y2* parameters. Users selecting the sfmr option should not insert any parameter for x1, x2, y1, and y2. In other words, the notation of the function is CNT(inp, wavelength, dx, dy, x1, x2, y1, y2, spatialFilter = 'sfr') or CNT(inp, wavelength, dx, dy, spatialFilter = 'sfmr'). This function is based on the proposed method by Kemper *et al.* [59]. We have implemented a search algorithm with nested for loops [53] to automatically find the best-reconstructed phase image (e.g., fully-compensate). |

power spectrum of the hologram of a transverse section of the head of a Drosophila melanogaster fly is shown to demonstrate that an off-axis setup in telecentric regime is used. After applying the three different approaches in this package, the reconstructed phase images are illustrated in Fig 5(C)–5(E).

Regarding the CNT function, a hologram of a *Drosophila melanogaster* fly recorded in non-telecentric regime has been used [30]. The parameters of the DHM system to record this sample are wavelength λ = 633 nm and a camera with a pixel size $dx = dy$ = 6.9 μm. The sample code is shown in Fig 6(A). The CNT function allows the spatial filtering of the object frequencies from the hologram spectrum using a rectangular mask defined by their pixel corners (e.g., *x1*, *x2*, *y1*, and *y2*) or manually drawn using a popup window. Before running the CNT function with the spatial filter option *sfr*, the user may need to select the pixels position (*x1*, *y1*) and (*x2*, *y2*) of the rectangular mask to filter the +1 term. For this task, one should compute the Fourier transform of the hologram (line 7). Fig 6(B) and 6(C) shows the Fourier transform and the binarized Fourier transform of the hologram. The latter display is recommended to select the positions of the rectangular mask, the red rectangle inside Fig 6(C). Line 11 runs the CNT function using the two possible filter options (e.g., *sfr* and *sfrm*). When running this function, the binarized phase image after compensating the interfering angle but without compensation of the quadratic phase factor is shown [Fig 6(D)]. Using this image, the user selects the

**(a)** **Example code for the compensation strategies in telecentric regime**

```
1- # import packages
2- from pyDHM import utilities
3- from pyDHM import phaseCompensation

4- #Load the hologram
5- inp = utilities.imageRead('hologram.jpg')

6- #Phase compensation using FRS, ERS, CFS
7- output = phaseCompensation.FRS(inp, True,
               0.633, 6.9, 6.9, 2, 10)
output = phaseCompensation.ERS(inp, True,
               0.633, 6.9, 6.9, 5, 0.2)
output = phaseCompensation.CFS(inp, 0.532, 2.6, 2.6)

8- #Display the phase reconstruction
9- phase = utilities.phase(output)
10- utilities.imageShow(phase, 'Phase reconstruction')
```

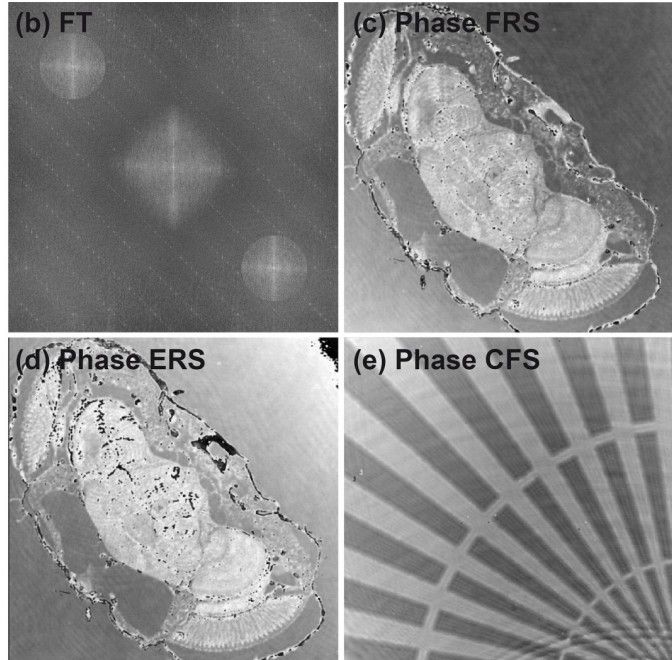

**Fig 5. Example of the fully-compensated phase reconstruction package for off-axis DHM holograms recorded in telecentric configuration.** (a) A sample code; (b) FT of a recorded hologram to show that the DHM system operates in off-axis and telecentric configuration. Panels (c)-(d) are the fully-compensated reconstructed phase images of a *Drosophila melanogaster* fly using FRS and ERS functions. Panel (e) shows the fully-compensated reconstructed phase image of a star target using the CFS function.

central pixel position (X_cent, Y_cent) of the quadratic phase mask. After closing this image, the CNT function asks to the user the values for this position: "*Enter the pixel position X_cent for the center of circular phase map on x axis,*" and "*Enter the pixel position Y_cent for the center of circular phase map on y axis*" for X_cent and Y_cent, respectively. The curvature of the quadratic phase factor is estimated by the size of the rectangular mask since the spreading of the +1 term and the curvature value are inversaly related [57]. When these values are introduced, the search for the reconstructed phase image starts, providing the optimal reconstructed phase image with minimum phase distortions [Fig 6(E)].

## 3.4 Package 4: Numerical propagation package

The final package in *pyDHM* is the numerical propagation package. This package contains the numerical propagation algorithms to compute the scalar complex diffractive wavefield at different propagation distances. The package is called by the following code line *from pyDHM import numerical Propagation*. We have implemented three different propagators: angular spectrum (*angularSpectrum*), the Fresnel transform (*fresnel*), and the Fresnel-Bluestein transform (*bluestein*). Table 4 shows the declaration statement and the parameters needed for each propagator. For example, the *angularSpectrum* and *fresnel* propagator functions have five parameters. *field* is the input complex wavefield to be propagated. The distance to propagate the input wavefield is represented by *z. wavelength* is the wavelength of the illumination source used to record the hologram. Finally, *dx* and *dy* are the pixel size for the input and output planes along the x- and y- directions. The *bluestein* propagator function has two additional parameters (e.g., *dxout* and *dyout*) related to the pixel size at the output plane.

**(a) Example code for the compensation strategy in non-telecentric regime**

```
1- # import packages
2- from pyDHM import utilities
3- from pyDHM import phaseCompensation

4- #Load the hologram
5- inp = utilities.imageRead('hologram.tif')

6- #FT of the hologram
7- ft_holo = utilities.FT(inp)
8- ft_holo = utilities.intensity(ft_holo, True)
9- utilities.imageShow(ft_holo, 'FT hologram')

10- #Phase compensation using CNT approach
11- output = phaseCompensation.CNT(inp, 0.633, 6.9,
              6.9,200, 287, 180, 267, spatialFilter = 'sfr')
output = phaseCompensation.CNT(inp, 0.633, 6.9,
              6.9, spatialFilter = 'sfmr')

12- #Display the phase reconstruction
13- phase = utilities.phase(output)
14- utilities.imageShow(phase, 'Phase reconstruction')
```

**Fig 6. Example of the CNF function for off-axis DHM holograms recorded in a non-telecentric configuration.** (a) A sample code; (b) FT of a recorded hologram, notice that the hologram operates in off-axis and non-telecentric configuration. Panel (c) is the binarized image of the FT to select the dimensions of the rectangular filter with parameters (x1, x2, y1, y2). Panel (d) corresponds to the binarized reconstructed phase image after compensating the interfering angle, where X_cent and Y_cent positions are marked. Finally, panel (e) shows the reconstructed phase image of a *Drosophila melanogaster* fly with minimum phase distortions.

We have evaluated the *angularSpectrum* propagator by numerically focusing the out-of-focus hologram of a USAF test target recorded in off-axis configuration [64]. The hologram was recorded using a wavelength of 633 nm. The camera with a pixel size $dx = dy = 6.9$ μm was located approximately 3 cm from the back focal plane of the TL. Fig 7(A) shows a sample code. The code starts with the import of the utility, and the numerical propagation packages; see lines 2–3 in Fig 7(A). In lines 5–6, the hologram is loaded and displayed [Fig 7(B)]. The Fourier Transform of the hologram is computing (line 8) and displayed (lines 9–10) in Fig 7(C). We have applied a circular mask (line 12) to filter the spatial frequencies of the object from the hologram spectrum [Fig 7(D)]. Line 14 calls the angular spectrum function. We have used three different values for the propagation distance ($z = 0$, 1, and 3.3 cm) to show the reconstructed intensity image (line 16–17) at different planes in Fig 7(E)–7(G). Inside each panel,

**Table 4. Available functions in the numerical propagation package.**

| Propagator | Specifications |
|---|---|
| **Angular spectrum** | angularSpectrum(field, z, wavelength, dx, dy) |
| | Function to propagate a complex distribution using the angular spectrum approach. |
| **Fresnel transform** | fresnel(field, z, wavelength, dx, dy) |
| | Function to propagate a complex distribution using the Fresnel Transform approach. |
| **Fresnel-Bluestein transform** | bluestein(field, z, wavelength, dx, dy, dxout, dyout) |
| | Function to propagate a complex distribution using the Fresnel-Bluestein transform approach. Note that the pixel size of the input (*dx*, *dy*) and the output (*dxout*, *dyout*) planes can be different in this method. |

**(a) Example code for propagation with angular spectrum**

```
1- # import packages
2- from pyDHM import utilities
3- from pyDHM import numericalPropagation

4- #Load the hologram
5- input= utilities.imageRead('hologram.tif')
6- utilities.imageShow(input, 'Out-of-focus Hologram')

7- #FT of the hologram
8- ft_holo = utilities.FT(input)
9- ft_holo = utilities.intensity(ft_holo, True)
10- utilities.imageShow(ft_holo, 'FT hologram')

11- #Circular spatial filter
12- filter = utilities.sfc(input, 160, 303, 276, True)

13- #Propagation using the angular spectrum
14- output = numericalPropagation.angularSpectrum(filter, z,
          0.633, 6.9, 6.9)

15- #Display the output field
16- intensity = utilities.intensity(output, False)
17- utilities.imageShow(intensity, 'Output field')
```

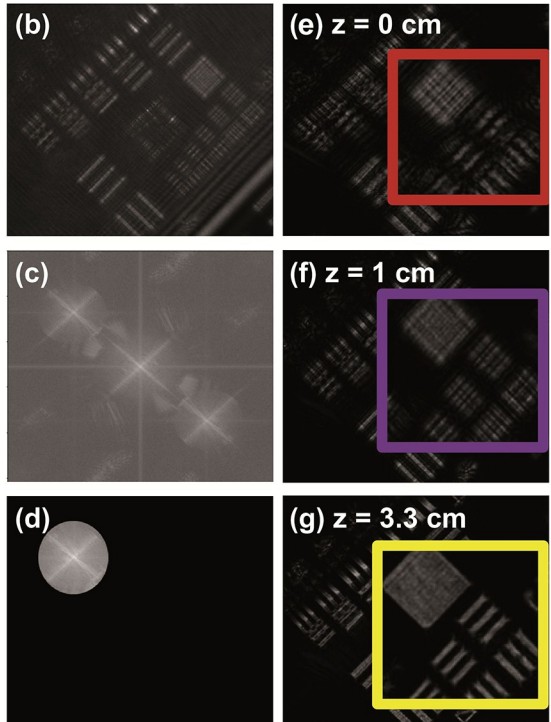

**Fig 7. Example of the angular spectrum approach to numerically focus a hologram using a circular spatial filter tool.** Panel (a) is the sample code. Panels (b)-(d) show the hologram (b) and its spectrum without (c) and with (d) a circular mask. Panels (e)-(g) are the reconstructed intensity images for three propagation distances ($z$).

the zoom-in rectangle areas highlight the effect of the propagation distance to focus the USAF target.

The performance of the Fresnel and Fresnel-Bluestein transform propagators has been validated by propagating an experimental hologram recorded using a Mach-Zehnder interferometer [61]. A hologram of a horse model is used for the Fresnel propagator, whereas a 1cm edge dice is used for the Fresnel-Bluestein propagator. The sample code for the *fresnel* and *bluestein* propagators is shown in Fig 8(A). Lines 2–3 are used to import the utility and numerical propagation packages. In lines 8–10, the computing and display of the Fourier transform of the hologram is implemented. In these examples, we have used a rectangle mask (line 12) to filter the hologram spectrum's object frequencies [Fig 8(B) and 8(E)]. Lines 14 and 15 calls the *fresnel* and the *bluestein* propagators, respectively. The parameters for the horse model hologram are $z$ = 45 cm, *wavelength* = 633 nm, and $dx = dy$ = 5 μm. Fig 8(C) and 8(D) are the reconstructed amplitude images without [Fig 8(C)] and with [Fig 8(D)] spatial filtering of the object frequencies in the hologram spectrum. The reconstruction parameters for the dice hologram are $z$ = 30 cm, $\lambda$ = 633 nm, and $dx = dy$ = 7.4 μm. The most important feature of the *bluestein* function (line 15) is that the output pixel sizes (*dxout* and *dyout*) are required as input parameters. Therefore, one can control the magnification of the reconstructed image using the Fresnel-Bluestein approach by modifying the value of these paramters. For this example, the output pixel sizes have been adjusted to 14.8 μm [Fig 8(F)], and 18.5 μm [Fig 8(G)]. These values provide an effective lateral magnification of 2× and 2.5× to the output size of the original hologram.

**(a)** **Example code for propagation with Fresnel and Fresnel-Bluestein transform**

```
1- # import packages
2- from pyDHM import utilities
3- from pyDHM import numericalPropagation

4- #Load the hologram
5- input= utilities.imageRead('hologram.bmp')
6- utilities.imageShow(input, 'Out-of-focus hologram')

7- #FT of the hologram
8- ft_holo = utilities.FT(input)
9- ft_holo = utilities.intensity(ft_holo, True)
10- utilities.imageShow(ft_holo, 'FT hologram')

11- #Rectangular spatial filter
12- filter = utilities.sfr(input, 280, 500, 150, 340, True)

13- #Propagation using Fresnel transforms
14- output = numericalPropagation.fresnel(input, -450,
              0.000633, 0.005, 0.005)
   output = numericalPropagation.bluestein(filter, 0.03, 633e-9,
              7.4e-6, 6.4e-6, 1.48e-5, 1.48e-5)

15- #Display the output field
16- intensity = utilities.intensity(output, False)
17- utilities.imageShow(intensity, 'Output field')
```

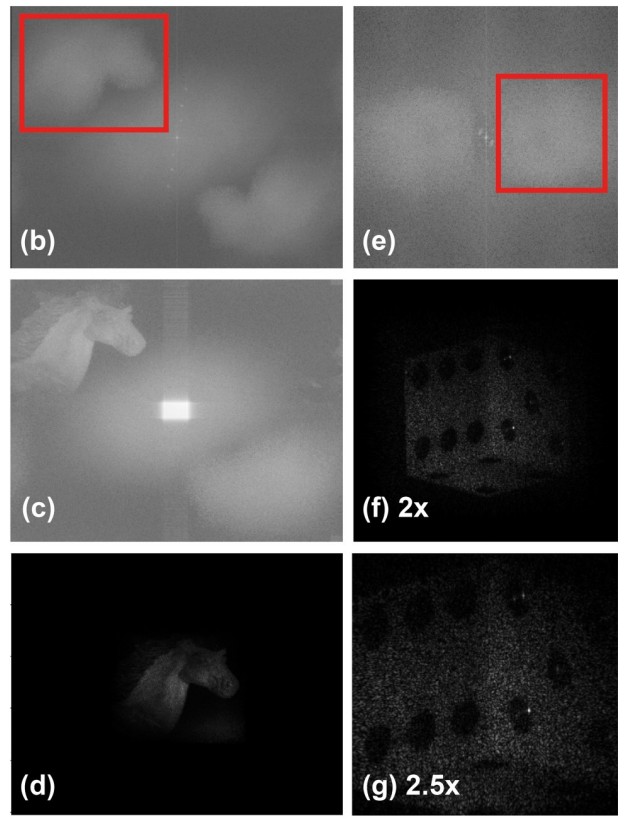

**Fig 8. Examples of the Fresnel and Fresnel-Bluestein propagation approaches to numerically focus holograms using a rectangular mask to filter the object frequencies from the hologram spectrum.** Panel (a) is the sample code. Panels (b) show the hologram spectrum (b) and the reconstructed intensity image without (c) and with (d) a rectangular mask of a horse model using the *fresnel* propagator. Panels (e)-(g) are the hologram spectrum (e), and the reconstructed intensity images after filtering the object frequencies from the hologram spectrum using two different magnifications [2× and 2.5× in panels (f) and (g), respectively] for a 1-cm edge dice using the *bluestein* propagator.

## 4. Conclusions

This work presents the pyDHM library, a Python library for the numerical processing of digital holograms registered in DHM systems. The library contains different computational implementations for: (1) reading and showing the complex distribution of a sample (e.g., utility package); (2) performing numerical propagations of complex wavefields to provide in-focus DH and DHM images (e.g., numerical propagation package); (3) reconstructing the phase distribution of samples in in-line and slightly off-axis DH and DHM systems using PS techniques (e.g., phase-shifting package); and (4) reconstructing phase images in single-shot off-axis DHM systems operating in telecentric and non-telecentric configuration using automatic methods to estimate the best digital reference wavefront (e.g., fully-compensated phase reconstruction package). We have presented a sample code for each function implemented in the pyDHM library and validated its performance using simulated or experimental images. The pyDHM library is posted publicly on GitHub [65, 66]. The GitHub repository includes a complete documentation of the functions implemented, sample codes, and troubleshooting guidelines for correctly using the library. To increase the applicability of this library in our community, the GitHub repository also includes simulated and experimental holograms and some instructional videos on how to install and use the library [67–69]. In future works, we will expand the codes within our library and reduce its processing time using GPU

implementations. Current implementations within the pyDHM library require that the users select the adequate reconstruction method based on the optical configuration of the DHM systems. Future work will focus on an automatic algorithm to reconstruct DHM holograms without prior knowledge of the DHM configuration (e.g., only hologram, source's wavelength, and sensor's pixel size). Because of the broad applicability of DHM systems in biology and medicine, we will create a graphical user interface (GUI) for the pyDHM library, aiming that users who lack coding skills and background in Optics and DHM could adopt this library. Such an app will allow the users to input a single hologram or a sequence/video of holograms, enabling the processing of the whole hologram series. Some reprocessing steps will be avoided, such as selecting the spatial filter mask to optimize such video processing. Our final goal is to expand such app further so that pyDHM can be used to analyze biological systems, including motility analysis for microorganism tracking and cell counting.

## Author Contributions

**Conceptualization:** Raul Castañeda, Carlos Trujillo, Ana Doblas.

**Funding acquisition:** Carlos Trujillo, Ana Doblas.

**Software:** Raul Castañeda, Carlos Trujillo.

**Supervision:** Carlos Trujillo, Ana Doblas.

**Validation:** Carlos Trujillo, Ana Doblas.

**Writing – original draft:** Raul Castañeda, Carlos Trujillo, Ana Doblas.

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
