## [Decision Letter · Decision Letter 0]

4 Aug 2022

PONE-D-22-18805pyDHM: A Python library for applications in Digital Holographic MicroscopyPLOS ONE

Dear Dr. Doblas,

Thank you for submitting your manuscript to PLOS ONE. After careful consideration, we feel that it has merit but does not fully meet PLOS ONE’s publication criteria as it currently stands. Therefore, we invite you to submit a revised version of the manuscript that addresses the points raised during the review process.

We look forward to receiving your revised manuscript.

Kind regards,

Luis Carretero, Ph.D

Academic Editor

PLOS ONE

Journal Requirements:

Reviewers' comments:

Reviewer's Responses to Questions

**Comments to the Author**

1. Is the manuscript technically sound, and do the data support the conclusions?

Reviewer #1: Partly

Reviewer #2: Yes

2. Has the statistical analysis been performed appropriately and rigorously? 

Reviewer #1: Yes

Reviewer #2: N/A

3. Have the authors made all data underlying the findings in their manuscript fully available?

Reviewer #1: Yes

Reviewer #2: Yes

4. Is the manuscript presented in an intelligible fashion and written in standard English?

Reviewer #1: Yes

Reviewer #2: Yes

5. Review Comments to the Author

Reviewer #1: The manuscript is clearly written. The authors conducted a thorough literature review in the field of digital holographic microscopy. Discussions on a broad range of DHM optical configurations and reconstruction methods are supported by appropriate references. The key innovation in this work is the proposal of the pyDHM library as a complete toolbox for numerical processing of holograms recorded from DHM systems of different configurations (on/slightly-off/off axis) with varying optical (tele centricity, wavelength, sensor properties) and acquisition parameters (focusing, phase-shifting intervals). However, there are several inconsistencies between the manuscript and the pyDHM library that require immediate corrections. Nevertheless, the pyDHM library provides a simplified approach in generalizing DHM numerical processing, which is beneficial for the field. I believe that the manuscript can be published in PLOS ONE as a research article, if the authors address the following comments.

1. As a coding/programing-orientated work, a thorough documentation for the pyDHM library should be composed and included on the GitHub repository for this work. The manuscript demonstrates sample codes for numerical processing, however, is not sufficient for facilitating troubleshooting. A detailed documentation that is standalone can immediately help readers and users trail the sample data, adopt this work to their workflow, troubleshoot, and contribute. The pyDHM library should also incorporate appropriate error messages to guide the users upon error.

2. It is understandable that the pyDHM library serves as the first toolbox library that supports multimodality DHM numerical processing. However, its useability is poor at current stage. Immediate future work should be aiming to improve the useability with for example, a graphical user interface. There is no doubt that the spatial filtering process of the +-1 orders in the FT of an off-axis hologram can be greatly simplified if performed in a graphical way.

3. In Fig. 6a, line 7 ‘hologram’ is not defined and an error is prompted when the sample code is run, possibly replacing by ‘inp’. Line 9 ‘ft_holo’ should be ‘ftholo’. It is recommended for the authors to doublecheck the variable names in sample codes throughout the manuscript.

4. In Fig. 6a, running line 11 – ‘output = phaseCompensation.CNT(inp, 0.633, 6.9, 6.9, 200, 287, 180, 267)’ in Python gives an error ‘TypeError: CNT() missing 3 required positional arguments: 'cur', 's', and 'step'’ and prevent further trails. The authors should conduct a thorough proof on the codes before publishing.

5. Currently, the pyDHM library only permits single hologram processing, which can be highly inefficient when users have a series of holograms or a video to process. To expand the temporal processing capability of the pyDHM library, the authors should provide a future plan on how this function can be achieved.

Reviewer #2: The manuscript is written clearly and understandably. An extensive literature review has been provided and delineation to the literature has been presented. In-depth knowledge of digital holographic microscopy and detailed explanations of the advantages and disadvantages of various techniques and algorithms are provided.

The authors contribute to community efforts by sharing the software publicly on Github. Code examples are available, which facilitates the introduction to the package.

The software is documented in the manuscript, however I would also suggest to add a documentation on github or dedicated software documentation services.

I believe the manuscript and the python package are useful contributions to the digital holographic microscopy community.

6. PLOS authors have the option to publish the peer review history of their article (what does this mean?). If published, this will include your full peer review and any attached files.

Reviewer #1: No

Reviewer #2: No

---

## [Author Response · Author response to Decision Letter 0]

21 Sep 2022

The authors thank the anonymous reviewer for reading the manuscript and for his/her effort in providing constructive criticism and positive feedback. A detailed reply to the reviewer's comments is provided below on the uploaded document named "Response to Reviewers". Our responses are written in blue font below each reviewer's comment, and the corresponding manuscript changes are provided in red font for clarity in the document "Revised Manuscript with Track Changes". We have numbered the reviewer's comments. In cases where numbers were not provided, we have numbered the key points asked so that each point could be addressed clearly.

---

## [Editor Report · Decision Letter 1]

26 Sep 2022

pyDHM: A Python library for applications in Digital Holographic Microscopy

PONE-D-22-18805R1

Dear Dr. Doblas,

We’re pleased to inform you that your manuscript has been judged scientifically suitable for publication and will be formally accepted for publication once it meets all outstanding technical requirements.

Kind regards,

Luis Carretero, Ph.D

Academic Editor

PLOS ONE

---

## [Editor Report · Acceptance letter]

28 Sep 2022

PONE-D-22-18805R1 

*pyDHM*: A Python library for applications in Digital Holographic Microscopy 

Dear Dr. DOBLAS:

I'm pleased to inform you that your manuscript has been deemed suitable for publication in PLOS ONE. Congratulations! Your manuscript is now with our production department. 

Kind regards, 

on behalf of

Dr Luis Carretero 

Academic Editor

PLOS ONE